# Survey of Clinical Practice Patterns of Korean Medicine Doctors for Anorexia in Children: A Preliminary Study for Clinical Practice Guidelines

**DOI:** 10.3390/children9091409

**Published:** 2022-09-17

**Authors:** Jihong Lee, Sun Haeng Lee, Jae Hyun Kim, Yong Seok Park, Sulgi Park, Gyu Tae Chang

**Affiliations:** 1Department of Korean Pediatrics, College of Korean Medicine, Daegu Haany University, 136 Sincheondong–ro, Suseong-gu, Daegu 42158, Korea; 2Department of Korean Pediatrics, College of Korean Medicine, Kyung Hee University, Kyung Hee University Medical Center, 23 Kyungheedae-ro, Dongdaemun-gu, Seoul 02447, Korea; 3Department of Clinical Korean Medicine, Graduate School, Kyung Hee University, 26 Kyungheedae-ro, Dongdaemun-gu, Seoul 02447, Korea; 4Department of Korean Pediatrics, College of Korean Medicine, Kyung Hee University, Kyung Hee University Hospital at Gangdong, 892 Dongnam-ro, Gangdong-gu, Seoul 05278, Korea

**Keywords:** anorexia, appetite, children, health surveys, traditional Korean medicine

## Abstract

Parents often have concerns regarding anorexia in their children and visiting medical institutions for the intervention of it. This study aimed to investigate the clinical practice patterns of Korean medicine doctors (KMDs) for anorexia in children using a web-based survey. A link to the questionnaire was sent via email to all KMDs that were affiliated with the Association of Korean Medicine. The questionnaire covered items on the sociodemographic characteristics and clinical characteristics related to Korean medicine (KM), such as diagnosis, treatment, awareness, safety, and effectiveness. Of 23,910 KMDs, 384 agreed to participate and complete the questionnaire. Anorexia in children was diagnosed mainly by clinical features (36.4%) and the pattern identification (PI) theory of ‘Qi, Blood, Fluid, Humor, and Organ system diagnoses’ (32.8%). The most frequently used PIs was ‘spleen-stomach qi deficiency’ (38.6%), which was followed by ‘spleen failure in transportation’ (23.3%), ‘stomach yin deficiency’ (15.5%), and ‘liver depression’ (14.2%). Herbal medicine (38.1%) was the primary KM treatment for anorexia, and the names of the most frequently prescribed herbal decoctions were Sogunjung-tang (16.5%), Hyangsayukgunja-tang (15.9%), and Bojungikgi-tang (13.9%). This study provides information on the existing clinical practice patterns of KMDs for anorexia in children. Based on this survey, the clinical practice guidelines will be developed.

## 1. Introduction

According to the Diagnostic Classification of Mental Health and Developmental Disorders of Infancy and Early Childhood (DC:0–3R), anorexia in children is characterized by (a) the refusal to eat sufficient amounts of food for more than 1 month, which is accompanied by growth retardation; (b) a lack of hunger and interest in food; and (c) no association of these symptoms with traumatic events or underlying medical conditions, and these criteria apply to children that are up to 3 years of age. [1]. Children require a relatively high energy intake due to their rapid growth and development. For this reason, many parents are concerned about anorexia in their children [2].

Anorexia in children is classified under feeding difficulties, which are divided into a limited appetite, selective intake, and the fear of feeding [3]. The signs and symptoms of feeding difficulties include prolonged mealtimes, stressed mealtimes, food refusal, a lack of adequate independent eating, or distractions to increase intake [2]. Feeding difficulties in children appear in a wide range of severities, from mild to severe, and the term is used indiscriminately along with others such as neophobia, picky eating, and feeding disorders in the literature [2]. Meanwhile, the DC:0-5 Diagnostic Classification of Mental Health and Developmental Disorders of Infancy and Early Childhood [4], which was published in 2016, uses the term ‘undereating disorders’, which refers to the child eating less than is expected for their age and exhibiting maladaptive dietary behavior [5]. Undereating disorders fall under the diagnosis of ‘unspecified feeding or eating disorder’ in the DSM-5, and they fall under ‘other eating disorder (F50.8)’ in the ICD-10 [6]. A diagnosis of ‘avoidant/restrictive food intake disorder (6B83 in ICD-11)’ can be made if an eating or feeding disorder results in a lack of adequate nutrition or energy, which adversely affects weight gain and height growth [7].

Twenty to forty percent of infants and toddlers have been reported to have eating and feeding problems [2,8,9,10,11]. In a cross-sectional study of pediatric patients who visited the Department of Pediatrics in the Hospital of Korean Medicine in the Republic of Korea, 1432 (30.6%) of the 4677 first-time patients visited the hospital for digestive system complaints, of which anorexia accounted for the largest proportion (1035 [72.2%]) [12]. In the Republic of Korea, the majority of children with anorexia mainly attend Korean medicine (KM) hospitals (68%) for counseling to determine their nutritional status and the cause of anorexia, followed by pediatrics (20%) and internal medicine (3%) hospitals [13]. East Asian medicine (including KM and Chinese medicine) treatment methods for anorexia involves treatment methods such as herbal medicine [14], acupuncture [15], and pediatric tuina (Chinese massage) [16]. On the other hand, the treatment for anorexia for children in Western medicine includes psychological treatment [17], zinc supplementation [18], and pharmacotherapy [19].

Clinical practice guidelines (CPGs) were developed with the objective of helping healthcare professionals, patients, and policymakers to make appropriate decisions regarding health issues [20]. Herbal medicines and acupuncture have shown significant effects among children who have anorexia [21,22,23], but in either the field of KM or Western medicine, CPGs have not been developed for anorexia in children in Korea. When it comes to treating children who have anorexia with KM, their diagnosis or treatment has not been standardized. Anorexia is recognized to be a major problem for young children under the age of three, but 7–27% of school-aged children also experience it [24]. In Korea, not only infants that are under 3 years old, but also children that are after school age are visiting KM institutions to improve their digestive problems, including that of anorexia [12]. Therefore, comprehensive CPGs for anorexia treatment that are targeting not only infants but also all children that are under the age of 20 are necessary. The CPGs need to reflect the current status of KM treatments, but an investigation into the current status of the treatment methods that are used by KM doctors (KMDs) has not been made. This study aimed to investigate the diagnosis and treatment methods that are currently used by KMDs through a survey, and to produce relevant CPGs.

## 2. Materials and Methods

### 2.1. Study Design

We conducted a cross-sectional web-based survey targeting the KMDs who provided KM treatment to anorexic children. The study was conducted in accordance with the guidelines of the Declaration of Helsinki, and it was approved by the Institutional Review Board of Kyung Hee University Korean Medicine Hospital at Gangdong, Korea on 8 October 2021 (KHNMCOH 2021-08-002-002).

### 2.2. Questionnaire Development

The questionnaires were developed by the researchers to investigate the patterns of KM for anorexia in children. The initial draft was created after reviewing information from comprehensive literature searches that were conducted by four KMDs. All of the four KMDs graduated from a 6-year university course in KM, and one researcher (CGT) is a professor at the KM university and a KM pediatrician with more than 20 years of clinical experience. The other (LJ) is a KM pediatrician with more than 10 years of experience. The other two KMDs (PYS and KJH) are residents that are specializing in KM pediatrics with more than 3 years of clinical experience.

Two extramural experts reviewed the initial draft and commented on the amendments. The extramural experts were KMDs with >10 years of clinical experience. Based on the opinions of the advisory committee, the researchers revised the questionnaire, which was a process that was confirmed in two research meetings. The final version of the questionnaire included items on (a) the sociodemographic status of the respondents (i.e., sex, age, years of clinical experience, affiliated institution, and subspecialty); (b) current status of diagnosis and KM treatment; (c) perception, safety, and effectiveness of KM treatment for anorexia in children; (d) the need for each CPG item. In the questionnaire, questions about pattern identification and the names of herbal medicines were based on the KM pediatric textbook [25]. The final version of the questionnaire is available in Appendix A.

### 2.3. Distribution and Collection of Questionnaires

We conducted a survey by targeting participants who satisfied the following conditions: KMDs who (1) graduated from the University of KM, (2) have KM licenses, (3) are members of the Association of Korean Medicine, and (4) had answered the questionnaire by 17 September. A link to the questionnaire was sent to the members (a total of 23,910) by email through the Association of Korean Medicine, to which all KMDs belong, regardless of age, sex, and region. The emails were sent on 7 and 14 September 2021, and the responses were collected on 17 September 2021. Moaform (https://surveyl.ink/) was used for the survey. All of the participants that voluntarily participated in the survey after being informed of the purpose of the study were guaranteed anonymity and signed informed consent to participate in the study. To increase the response rate, we provide incentives through the means of prize draws.

### 2.4. Statistical Analyses

A statistical analysis of the research data was performed using Excel version 2016 (Microsoft Corporation, Redmond, WA, USA) and SPSS (IBM Corporation, Armonk, NY, USA). The results were reported using descriptive statistics and a frequency analysis was performed for each questionnaire item. The items that allowed multiple responses were analyzed using a multiple response analysis. To compare by categorical data of clinical experience of the respondents, we analyzed the data with the Chi-square test. *p*-values < 0.05 were considered statistically significant.

## 3. Results

### 3.1. Demographic Characteristics of the Respondents

Of the 23,910 KMDs to whom the email was sent, 738 accessed the link and 385 responded to the survey. Except for one person who did not agree to participate, 384 KMDs completed the questionnaire, and the response rate was 1.61%. The demographic characteristics of the respondents are listed in Table 1 and are summarized as follows: 72.4% of the respondents were male, and the most common age groups were 30–39 years (35.9%) and 40–49 years (35.4%). Nearly a third of the respondents had 10–19 years of clinical experience, and 28.4% had 5–9 years of experience. Of the respondents, 66.1% worked in a primary KM clinic and 69.3% were general practitioners without specialized training. Most of the respondents (80.5%) answered that they had encountered fewer than five first-time pediatric patients with anorexia on average in the past month. Regarding the average treatment duration, 52.3% responded that it was 1–3 months. The predominant age groups of the children with anorexia treated by the respondents were of preschool age (1–7 years) (38.2%) and early elementary school age (7–9 years) (35.1%) (multiple responses were allowed).

### 3.2. Diagnosis Used for Children with Anorexia

Children with anorexia were largely diagnosed based on clinical features (36.4%) and the pattern identification (PI) theory of ‘Qi, Blood, Fluid, Humor, and Organ System’ using KM textbooks (32.8%) (Table 2). Of the titles of the PI, based on the KM textbook, the most frequently used title was ‘spleen-stomach qi deficiency’ (38.6%), which was followed by ‘spleen failure in transportation’ (23.3%), ‘stomach yin deficiency’ (15.5%), and ‘liver depression’ (14.2%). The most commonly used diagnostic equipment for evaluating children with anorexia were height and weight measurements (43.6%), which was followed by a bioelectrical impedance analysis (20.7%). The diagnostic equipment question allowed for multiple responses.

Currently, the diagnostic criteria for anorexia in children [1] are only applicable to children that are aged 0–3 years and they consisted of three items including “the child refuses to eat a sufficient amount of food for more than one month and growth is delayed”. Moreover, there are no definitive diagnostic criteria for children that are over three years of age. When asked whether the respondents agreed with the applicability of this criteria to all age groups, 77.3% answered that they agreed.

### 3.3. KM Treatments for Children with Anorexia (Intervention)

Among the KM treatments for the children with anorexia, herbal medicine (38.1%) was investigated as the most commonly used treatment, with acupuncture (21.5%), moxibustion (15.6%), and dietetic therapy (9.8%) being the next most used ones. Regarding the type of herbal medicines that are used, compound herbal decoctions were the most frequently used ones (57.1%). Herbal decoctions were usually prescribed in the following order: Sogunjung-tang (16.5%), Hyangsayukgunja-tang (15.9%), Bojungikgi-tang (13.9%), and Samchulgeonbi-tang (11.9%). As single herbs, Atractylodis Rhizoma Alba (18.5%), Citri Pericarpium (13.1%), and Crataegii Fructus (12.1%) were found to be frequently used in the treatment of anorexia in children. The duration of the herbal medicine treatment was 1–3 months in most of the cases (58.1%).

The style of acupuncture that was primarily used was meridian point acupuncture (45.7%); the main acupuncture points were ST36 (25%) and CV12 (20.9%). CV12 was used by 54.5% of the respondents as an acupoint for moxibustion. Regarding dietetic therapy, 51.7% of the respondents answered that dietetic therapy that was based on KM theory was used, while dietetic therapy that was based on nutritional science was used by 45.8% of the respondents. Multiple responses were allowed for all of the questions on the KM treatments, and the details of the items are presented in Table 3.

When analyzing the usage rates of major KM treatment methods by the clinical experiences of the respondents, a statistically significant difference was found in the utilization rates of the treatments between the different experience groups, with the *p*-value of herbal medicine showing 0.016 and that of acupuncture being < 0.001. In the case of moxibustion, there was no difference in the utilization rate between the five groups (*p* > 0.05) (Table 4).

### 3.4. Perception

When the KMDs answered a question on a Likert 5-point scale regarding the overall effect of KM treatment on the children’s anorexia as based on their treatment experience, the average score was 4.18 ± 0.68 (1 = not at all effective, 2 = mostly not effective, 3 = moderate, 4 = mostly effective, and 5 = very effective). Regarding the advantages of KM treatment for anorexia, 32.3% of the respondents answered that it was effective, 27% stated that it was a fundamental treatment for strengthening digestive function, and 26.1% reported that it had fewer side effects than other treatments. Regarding the points that needed to be supplemented to improve KM treatment for anorexia in children, 34.6% answered that one is the cost of treatment, 29.7% answered that another is the availability of treatment information, and 27.1% answered that another is the convenience of treatment (Table 5).

### 3.5. Safety and Effectiveness

The most frequently answered duration for evaluating the effectiveness of treatments for anorexia in children was 1 month (42.7%), which was followed by 3 months (38.8%). The effectiveness evaluation index most commonly used was in the following order: the amount of food intake (28.8%), weight (21.8%), improvement of overall physical condition (20.5%), and height (14%).

A safety evaluation was conducted every 1 month by 51% of the respondents, which was followed by every 3 months by 30.7% of the respondents. Regarding the safety evaluation, the KMDs responded that they considered changes in the child’s overall physical condition (37.6%), evaluation of adverse reactions (35.9%), and changes in vital signs (20.1%) as indicators of their safety.

Seven percent of the respondents answered that they encountered children who experienced an adverse reaction during the herbal medicine treatment of anorexia, and 1.6% responded that they encountered children who experienced an adverse reaction during acupuncture or moxibustion. Fourteen KMDs reported that the adverse reaction that occurred during the herbal medicine treatment was diarrhea, while four KMDs reported a skin rash or digestive complaints. Two KMDs responded that the adverse reaction that was experienced by the children was stress due to the fear of acupuncture. All of the six KMDs who reported an occurrence of an adverse reaction to moxibustion stated that these were burns (Table 6).

### 3.6. Information to Be Included in the CPGs

When they were asked about what they believed was the most pertinent information that was required in order to effectively treat anorexia in children, with up to three responses being allowed, the following were the most common responses (in order): evaluation method (22.3%), diagnostic criteria (20.1%), differential diagnosis (17%), and management method (15.1%). Table 7 shows the results of the content that should be included in the CPGs for anorexia in children for each item.

## 4. Discussion

We conducted a web-based survey on the current status of the treatment of children with anorexia with the aim of developing CPGs for pediatric patients undergoing KM treatment for anorexia. The questionnaire that was developed by the research team was designed by considering the opinions of experts and by conducting a comprehensive literature review, through which it was possible to obtain useful information on the diagnosis, treatment methods, awareness, safety, and effectiveness of KM for anorexia.

When they were surveyed about the methods that are used to diagnose children with anorexia, most respondents answered that they diagnosed them using clinical characteristics, while the KM textbook-based pattern identification of ‘Qi, Blood, Fluid, Humor, and Organ System’ was the second most frequently used method. In the case of a diagnosis that is based on clinical features, it may be difficult to objectively assess because it is subjectively determined by the physician, so the diagnosis may differ between KMDs and the findings may be influenced by the patient’s and the caregiver’s disposition. Therefore, clearer diagnostic criteria must be prepared, and an awareness and education of them are needed to allow clinicians to implement these criteria well. Regarding the PIs of ‘Qi, Blood, Fluid, Humor, and Organ System’ for describe anorexia in children, factors such as ‘spleen-stomach qi deficiency’, ‘spleen failure in transportation’, ‘stomach yin deficiency’, and ‘liver depression’ were frequently reported. In East Asian medicine, five organs of the digestive system, including the spleen and stomach, are frequently used to describe digestive disorders. In addition, the PI ‘Liver Qi stagnation and Spleen deficiency’ is a common syndrome pattern that is observed clinically in gastrointestinal symptoms, which occurs due to excessive psychological pressure. It is caused by liver dysfunction, which controls the flow of the qi, and spleen dysfunction, which controls digestion and transport. This emotional imbalance can cause digestive symptoms, including anorexia, bloating, abdominal pain, and diarrhea [25]. Liver qi stagnation and spleen deficiency syndrome may be important causes of anorexia in children, and it is necessary to explore these aspects when treating anorexia in children.

The diagnostic criteria for infantile anorexia in the DC: 0-3R are classified under ‘Feeding Behavior Disorders’ in the first 3 years of their life, and the corresponding anorexia criteria for children and adolescents that are over 3 years of age have not been clearly established. The DSM-5 [7] defines any form of anorexia that is not anorexia nervosa as an ‘unspecified feeding or eating disorder’. Therefore, we asked the KMDs for their opinion on extending the above criteria to all children that are aged 0–19 years. A total of 297 respondents (77.3%) answered that they agreed, 54 (14.1%) did not agree, and 33 (8.6%) said they were unsure. When the KMDs who responded that they did not agree were asked for the reason for this, six said it was difficult to apply the same method to all age groups because the cause of anorexia differs according to age. In addition, six KMDs suggested that the above criteria are applicable prior to the development of secondary sexual characteristics, while six KMDs stated that the adult criteria for anorexia can be applied after puberty. Based on the opinions that have been presented in this survey, additional opinions must be collected and confirmed through in-depth interviews or Delphi surveys with panels of KM pediatric specialists.

Furthermore, in the revised DC:0-5, the term ‘undereating disorder’ is used instead of ‘infantile anorexia’. The diagnostic criterion is that “the child continues to eat less than expected for his or her age and exhibits maladaptive dietary behavior”, which does not include factors like weight loss or failure to gain weight [5]. In the process of achieving an expert consensus on the diagnosis of anorexia in children, it is necessary to collect opinions by referring to the section on the ‘undereating disorder’.

The KM treatment for anorexia in children was mainly through the use of herbal medicine (38.1%) and among these herbal medicines, herbal decoction (57.1%) was primarily prescribed. When we analyzed the groups according to the length of their clinical experience, there was a statistically significant difference in the use of herbal medicines (*p* = 0.016) and acupuncture (*p* < 0.001) to treat anorexia in children (Chi-square test). Sogunjung-tang, Hyangsayukgunja-tang, Bojungikgi-tang, and Samchulgeonbi-tang were frequently prescribed decoctions. Sogunjung-tang has been reported to be effective in treating digestive system complaints, such as functional gastrointestinal disorders and psychosomatic disorders in patients with gastrointestinal diseases in East Asian medicine [26,27]. In the case of Hyangsayukgunja-tang, its clinical efficacy has been reported in pediatric abdominal pain patients [28], and a meta-analysis of randomized controlled trials has demonstrated that it was more effective than prokinetic drugs were in treating functional dyspepsia [29]. Moreover, experimental studies have shown that it improves appetite [30], and damages the gastric mucosa by affecting the gastrointestinal hormones [31]. Bojungikgi-tang has been used for treating gastrointestinal diseases, such as functional constipation [32] and intestinal mucositis [33], and it has been found to increase the levels of the gastrointestinal hormones and stimulate gastrointestinal motility in experimental studies [34,35,36].

In the treatment experiences of the KMDs, when the overall effectiveness of the KM treatment for children with anorexia was assessed on a 5-point Likert scale, ‘most effective (4 points)’ was predominantly selected (54.4%), which was followed by ‘very effective (5 points)’ (32.6%). Moreover, considering that the respondents most commonly selected ‘effective’ as an advantage of KM treatment (32.3%), it appears that the KMDs are satisfied with KM treatment for anorexia, from the practitioner’s point of view. Other advantages of KM treatment are that it is a “fundamental treatment that strengthens digestive function” and that it has “fewer adverse effects.” Conversely, treatment cost, information on treatment, and treatment convenience were suggested as aspects that need to be improved in the KM treatment for anorexia. Regarding the cost of treatment, the average treatment cost per treatment for a child with anorexia (copay) as of last year was 10,000–20,000 won (46.9%) in most cases, and regarding the average treatment duration, 52.3% of the respondents answered “1–3 months.” Taking this into account, it is believed that such a cost can act as a burden on the patient’s caregivers. To improve the economic feasibility of this treatment, studies on the efficacy and safety of KM treatment for anorexia in children, as well as an economic evaluation of it, should be conducted. In addition, the use of various promotional communications to raise awareness for KM treatment and ways to increase children’s compliance with herbal medicines or acupuncture to improve treatment convenience should be explored. Furthermore, it is necessary to gather opinions from children who are receiving treatment or their caregivers on the pros and cons of such treatments.

According to the survey, the KMDs responded that they evaluated the amount of food that is consumed, weight, and an improvement in the general condition as efficient indicators for the treatment of anorexia in children. All these items differ in terms of diagnostic methods that are used and include aspects that may be subject to an arbitrary interpretation by the evaluator. Therefore, specific and objective measurements and interpretation methods were required for each item.

Most of the KMDs (90.1%) responded that they did not encounter children who experienced any adverse reactions during herbal medicine treatments. The adverse reactions that occurred during the herbal medicine treatments were mainly digestive or skin complaints, such as diarrhea and a rash. Therefore, it is necessary to monitor these two areas with particular care during herbal medicine treatment. The percentage of the KMDs who responded that they encountered patients who underwent adverse reactions during acupuncture and moxibustion treatment was 1.6%. The adverse reaction to acupuncture that was reported by two KMDs was the stress due to the fear of the acupuncture treatment. Therefore, it is necessary to reduce the pain that occurs during acupuncture in pediatric patients, and skilled KMDs are needed to decrease the amount of tension in this patient group. Six KMDs who reported adverse reactions to moxibustion stated that these were burns. Hence, moxibustion treatment must be performed carefully by a trained medical personnel.

This study has several limitations. First, the response rate to the questionnaire was low. As the email was sent to all KMDs that were registered with the Association of Korean Medicine, there may have been invalid email addresses, and some KMDs may not have opened the email at all. Therefore, the clinical status or opinions of the KMDs who were not contacted by email were not reflected. Second, because the effectiveness and safety of the questionnaire were reported from the perspective of the KMDs, a prospective study using actual patient data is needed to obtain more objective findings. In addition, to supplement the cross-sectional results of the email survey, an in-depth interviews with KMDs, pediatric patients, or caregivers who have received KM treatment can be conducted. A third limitation is our use of a non-standardized questionnaire. However, to account for this, KM pediatric specialists participated in designing the questionnaire, and revisions were made that were based on the opinions of a panel of extramural KM experts.

This is the first survey on the clinical practice patterns among KMDs for anorexia in children. In this cross-sectional study, only a fraction of all KMDs completed the response, but it was possible to provide a general overview on the treatment of anorexia in children by KMDs, including those from various working regions, workplaces, of different ages, and with different amounts of clinical experience. Based on this fact-finding survey of KMDs, KM CPGs for anorexia in children were drafted and certified by the Association of Pediatrics of Korean Medicine, and they are expected to be completed in 2022. The researchers collected the existing research results and confirmed the level of evidence that was needed and made recommendations for KM treatment; based on the results that were obtained, a systematic review of herbal medicine was published [14].

The CPGs, which include the medical status of the KMDs, can provide essential information on the efficacy and safety of KM treatments to patients and caregivers of KMDs that treat anorexia in children, and can help KMDs to make evidence-based medical decisions. Through this, the confidence in KM services can be increased, and the degree of patient satisfaction can be improved.

## 5. Conclusions

This study provides information on the current KMDs’ clinical practice patterns for anorexia in children. Based on this survey, which contains information on diagnosis and treatment, CPGs will be developed. The use of evidence-based CPGs will help medical professionals, patients, and caregivers to make rational medical decisions.

## Figures and Tables

**Table 1 children-09-01409-t001:** Sociodemographic characteristics of the surveyed Korean medicine doctors (N = 384).

Factors		N (%)
Age (years)	20–29	27 (7)
	30–39	138 (35.9)
	40–49	136 (35.4)
	50–59	66 (17.2)
	≥60	17 (4.4)
Sex	Male	278 (72.4)
	Female	106 (27.6)


Clinical experience (years)	≤4	57 (14.8)
	5–9	109 (28.4)
	10–19	118 (30.7)
	20–29	76 (19.8)
	≥30	24 (6.3)
Place of work	Seoul	129 (33.6)
	Gyeonggi	91 (23.7)
	Busan	20 (5.2)
	Daegu	18 (4.7)
Special training	No	266 (69.3)
	Yes	118 (30.7)
Name of the academic society for specialist training (if applicable)	The Society of Internal Korean Medicine	37 (31.4)
	Korean Acupuncture and Moxibustion Medicine Society	23 (19.5)
	The Society of Korean Medicine Obstetrics and Gynecology	18 (15.3)
	The Society of Korean Medicine Pediatrics	10 (8.5)
	The Society of Korean Medicine Rehabilitation	10 (8.5)
	The Society of Korean Medicine Ophthalmology, Otolaryngology & Dermatology	9 (7.6)
	The Society of Sasang Constitutional Medicine	6 (5.1)
	The Society of Korean Medicine Neuropsychiatry	5 (4.2)
Affiliated medical institution	Primary KM clinic	254 (66.1)
	University-affiliated KM hospital	36 (9.4)
	KM hospital (not a university affiliated hospital)	36 (9.4)
	Specialized KM clinic (KM clinic providing specialized treatment for specific diseases)	28 (7.3)
Monthly average number of first-time pediatric patients with anorexia that were encountered in the last month	≤5	309 (80.5)
	6–10	53 (13.8)
	11–20	13 (3.4)
	21–30	4 (1.0)
	31–40	1 (0.3)
	≥41	4 (1.0)
Average treatment period for children with anorexia in the last year	<1 month	120 (31.3)
	≥1 month to <3 months	201 (52.3)
	≥3 months to <6 months	36 (9.4)
	≥6 months to <1 year	21 (5.5)
	≥1 year to <3 years	4 (1.0)
	≥3 years	2 (0.5)
Average treatment cost per treatment for children with anorexia in the last year (copay) (Won) (If an herbal decoction is included, it is calculated by dividing the total dose by the number of treatment days)	<5000	9 (2.3)
	≥5000 to <10,000	45 (11.7)
	≥10,000 to <20,000	180 (46.9)
	≥20,000 to <50,000	72 (18.8)
	≥50,000 to <100,000	21 (5.5)
	≥100,000	57 (14.8)
Age range of the children with anorexia that were being treated (multiple responses allowed)	Infants (≥1 month to <1 year)	17 (2.4)
	Preschool children (≥1 year to <7 years)	267 (38.2)
	Children in early elementary school (7–9 years)	245 (35.1)
	Children in the upper grade of elementary school (10–12 years)	104 (14.9)
	Middle school students (13–15 years)	40 (5.7)
	High school students and above (≥16 years)	26 (3.7)

KM: Korean Medicine.

**Table 2 children-09-01409-t002:** Diagnosis of anorexia in children (N = 384).

Factors		N (%)
Diagnostic methods used for children with anorexia ^1^	Diagnosis based on the clinical features	288 (36.4)
	Pattern identification—Qi, Blood, Fluid, Humor, and Organ system diagnosis based on KM textbooks	260 (32.8)
	Pattern identification—Sasang constitutional diagnosis	101 (12.8)
	Pattern identification—Six-Meridian pattern identification based on “Cold Damage Medicine”	30 (3.8)
	KM diagnostic device	29 (3.7)
	Pattern identification—Hyungsang Constitutional medicine	26 (3.3)
	Korean Children’s Eating Behavior Questionnaire	24 (3.0)
	Blood and urine test	19 (2.4)
	Pattern identification—Eight Constitution medicine	11 (1.4)
Title of frequently used pattern identification ^1^	Spleen-stomach qi deficiency	244 (38.6)
	Spleen failure in transportation	147 (23.3)
	Stomach yin deficiency	98 (15.5)
	Liver depression	90 (14.2)
	Milk and food damage	43 (6.8)
	Others	10 (1.6)
Diagnostic equipment used for children with anorexia ^1^	Height and weight measurement	281 (43.6)
	Bioelectrical impedance analysis (InBody)	133 (20.7)
	None	73 (11.3)
	KM diagnostic device—Heart Rate Variability	59 (9.2)
	Blood and Urine Test	22 (3.4)
	KM diagnostic device—Electro Pulse Graph	19 (3.0)
	KM diagnostic device—Yangdorak Diagnosis	18 (2.8)
	Digital Infrared Thermal Imaging	15 (2.3)
	KM diagnostic device—Tongue Diagnosis	9 (1.4)
	KM diagnostic device—Constitutional Diagnosis	5 (0.8)
	KM diagnostic device—Iris Diagnosis	4 (0.6)
	Others	6 (0.9)
Do you agree that the diagnostic criteria of anorexia for children aged 0–3 years should be applicable to all children (0–19 years)?	Yes	297 (77.3)
	No	54 (14.1)
	I do not know	33 (8.6)

KM: Korean Medicine; ^1^ multiple responses were allowed.

**Table 3 children-09-01409-t003:** Frequently used Korean medicine treatments for children with anorexia (N = 384).

Factors		N (%)
Treatment ^1^	Herbal medicine	375 (38.1)
	Acupuncture	211 (21.5)
	Moxibustion	153 (15.6)
	Dietetic therapy	96 (9.8)
	Cupping therapy	58 (5.9)
	Electroacupuncture	39 (4.0)
	Manipulation/Exercise therapy	37 (3.8)
	Pharmacopuncture	13 (1.3)
Type of herbal medicine formulation ^1,2^	Compound herbal decoction	363 (57.1)
	Soft extract covered by insurance	57 (9.0)
	Mixture of soluble granules covered by insurance	56 (8.8)
	Soft extract not covered by insurance	43 (6.8)
	Mixture of soluble granules not covered by insurance	34 (5.3)
	Distillate of compound herbal decoction	32 (5.0)
	Powder preparation	25 (3.9)
	Pill preparation	24 (3.8)
Name of herbal medicine ^1,2^	Sogunjung-tang	223 (16.5)
	Hyangsayukgunja-tang	214 (15.9)
	Bojungikgi-tang	187 (13.9)
	Samchulgeonbi-tang	161 (11.9)
	Hyangsayangwi-tang	130 (9.6)
	Insamyangwi-tang	90 (6.7)
	Jeonssiigong-san	84 (6.2)
	Samryeongbaekchul-san	81 (6.0)
	Bihwa-eum	37 (2.7)
	Bohwa-hwan	36 (2.7)
	Haeulgeonbi-tang	12 (0.9)
	Sosik-hwan	12 (0.9)
Single herb ^1,2^	Atractylodis Rhizoma Alba	321 (18.5)
	Citri Pericarpium	228 (13.1)
	Crataegii Fructus	210 (12.1)
	Hordei Fructus Germiniatus	201 (11.6)
	Massa Medicata Fermentata	199 (11.4)
	Glycyrrhizae Radix	167 (9.6)
	Poria	161 (9.3)
	Atractylodis Rhizoma	123 (7.1)
	Galli Stomachichum Corium	48 (2.8)
	Other—Ginseng Radix	11 (0.6)
	Other—Cervi Pantotrichum Cornu	9 (0.5)
Herbal medicine treatment duration ^2^	<1 month	70 (18.7)
	≥1 month to <3 months	218 (58.1)
	≥3 months to <6 months	73 (19.5)
	≥6 months to <1 year	10 (2.7)
	≥1 year to <3 years	3 (0.8)
	≥3 years	1 (0.3)
Style of acupuncture ^1,2^	Meridian points acupuncture	169 (45.7)
	Dermal needle	48 (13.0)
	Intradermal needle	47 (12.7)
	Sa-am acupuncture therapy	43 (11.6)
Acupoints used for acupuncture ^1,2^	ST36	172 (25.0)
	CV12	144 (20.9)
	PC6	96 (13.9)
	ST25	54 (7.8)
	SP6	43 (6.2)
	CV4	42 (6.1)
Acupoints used for moxibustion ^1,2^	CV12	120 (54.5)
	CV4	29 (13.2)
	ST36	21 (9.5)
	ST25	11 (5.0)
Type of dietetic therapy ^1,2^	Dietetic therapy based on KM theory	62 (51.7)
	Dietetic therapy based on nutritional science	55 (45.8)

KM: Korean Medicine; ^1^ multiple responses were allowed; ^2^ if applicable.

**Table 4 children-09-01409-t004:** Utilization patterns of main treatment methods by clinical experience (N = 384).

	Clinical Experience					
Treatment	≤4 Years (n = 57) (n (%))	5–9 Years (n = 109) (n (%))	10–19 Years (n = 118) (n (%))	20–29 Years (n = 76) (n (%))	≥30 Years (n = 24) (n (%))	*p*-Value (Χ^2^ Test)
Herbal medicine	54 (94.7)	106 (97.2)	115 (97.5)	76 (100)	24 (100)	0.016
Acupuncture	37 (64.9)	71 (65.1)	65 (55.1)	20 (26.3)	18 (75)	<0.001
Moxibustion	26 (45.6)	52 (47.7)	50 (42.4)	9 (11.8)	16 (66.7)	0.068

**Table 5 children-09-01409-t005:** Perception of Korean medicine in the treatment of children with anorexia (N = 384).

Statements		N (%)
1. From your treatment experience, what is the overall effect of KM treatment on anorexia in children?	Mostly effective	209 (54.4)
	Very effective	125 (32.6)
	Moderate	47 (12.2)
	Mostly not effective	2 (0.5)
	Not at all effective	1 (0.3)
2. What are the advantages of KM in the treatment of anorexia in children? ^1^	Effective	305 (32.3)
	Fundamental treatment to strengthen the digestive function	255 (27.0)
	Fewer side effects	246 (26.1)
	More helpful than other treatment methods	112 (11.9)
	Economical	24 (2.5)
3. What needs to be supplemented in KM for treating anorexia in children? ^1^	Treatment cost	265 (34.6)
	Treatment information (promotion and awareness)	227 (29.7)
	Convenience of treatment (herbal medicine and acupuncture)	207 (27.1)
	Duration of treatment	39 (5.1)

KM: Korean Medicine; ^1^ multiple responses allowed.

**Table 6 children-09-01409-t006:** Safety and effectiveness of Korean medicine treatments in children with anorexia (N = 384).

Factors		N (%)
Evaluation period of effectiveness in the treatment of anorexia in children	1 month	164 (42.7)
	3 months	149 (38.8)
	6 months	54 (14.1)
	1 year	11 (2.9)
	Others—2 weeks	5 (1.3)
Effectiveness evaluation index in the treatment of anorexia in children ^1^	Amount of food	340 (28.8)
	Weight	257 (21.8)
	Improvement in general condition of the child	242 (20.5)
	Height	165 (14.0)
	Body mass index	89 (7.5)
	Numeric rating scale	85 (7.2)
Safety evaluation period in the treatment of anorexia in children	1 month	196 (51.0)
	3 months	118 (30.7)
	6 months	53 (13.8)
	1 year	7 (1.8)
	Others—2 weeks	5 (1.3)
Safety evaluation index in the treatment of anorexia in children ^1^	Changes in the child’s general condition	296 (37.6)
	Evaluation of adverse reactions	283 (35.9)
	Changes in vital signs	158 (20.1)
	Blood test	39 (4.9)
	Urine test	12 (1.5)
Are there cases of adverse reactions during herbal medicine treatment for anorexia in children?	No	346 (90.1)
	Yes	27 (7.0)
	No herbal medicine treatment administered for children with anorexia	11 (2.9)
Are there cases of adverse reactions during acupuncture treatment for anorexia in children?	No	265 (69.0)
	Yes	6 (1.6)
	No acupuncture treatment administered for children with anorexia	113 (29.4)
Are there cases of adverse reactions during moxibustion treatment for anorexia in children?	No	205 (53.4)
	Yes	6 (1.6)
	No moxibustion treatment administered for children with anorexia	173 (45.1)

KM: Korean Medicine; ^1^ multiple responses were allowed.

**Table 7 children-09-01409-t007:** Content to be included in the clinical practice guidelines for anorexia treatment in children (N = 384).

Factors		N (%)
What information would you like to know more about treating anorexia in children? (up to three multiple responses allowed)	Evaluation method	257 (22.3)
	Diagnostic criteria	231 (20.1)
	Differential diagnosis	196 (17)
	Management	174 (15.1)
	Western medical treatments	157 (13.6)
	Treatment with Korean medicine	137 (11.9)
Diagnostic criteria that should be included in the development of CPGs ^1^	Diagnostic criteria for Korean medicine (pattern identification)	329 (54.2)
	Diagnostic criteria for Western medicine	277 (45.6)
	Other—Evaluation tool for objectively assessing amount of food/defecation	1 (0.2)
Evaluation method that should be included in the development of CPGs ^1^	Questionnaire used to assess symptom severity	284 (39.7)
	Questionnaire for pattern identification	236 (33)
	Food/symptom diary	196 (27.4)
Western medical treatment that should be included in the development of CPGs ^1^	Common adverse events and remedies	316 (53)
	Efficacy/mechanism of individual Western medicines	277 (46.5)
Korean medicine treatment methods that should be included in the development of CPGs ^1^	Herbal medicine	366 (27.2)
	Acupuncture	243 (18)
	Dietetic therapy	203 (15.1)
	Moxibustion	189 (14)
	Manipulation/Exercise therapy	136 (10.1)
Differential diagnosis that should be included in the development of CPGs ^1^	Specific information on similar diseases	288 (50.6)
	Information on the types of similar diseases	280 (49.2)

CPGs: clinical practice guidelines; ^1^ multiple responses were allowed.

## Data Availability

The data presented in this study are available upon request from the corresponding author.

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
