# Peer review of "Survey of Clinical Practice Patterns of Korean Medicine Doctors for Anorexia in Children: A Preliminary Study for Clinical Practice Guidelines"

_children, 2022, doi:10.3390/children9091409_

Round 1
Reviewer 1 Report
Thank you very much for the opportunity to review your manuscript. The topic seems interesting and relevant to clinical practice in terms of eating disorders. Despite the logical layout and interesting conclusions, I have a few comments for the authors:
1. it is not clear what the purpose of the study is. Of course, I can see the purpose at the end of the theoretical introduction, but it needs to be detailed and provide hypotheses (research questions), and the introduction itself does not clearly present the research gap. I suggest before the objective to specify the assumptions of the study being conducted.
2. please establish the criteria for qualifying the participants of the study.
3. I encourage you to expand statistical analyses to include the use of inference with statistical tests and determine the level of probability (p-value).
4. The methodology lacks mention of how many study participants there were - also refers to point 2 of this review, as this also refers to the inclusion and exclusion criteria of the study.
5. In the titles of the tables, the abundance (N valid) should be given.
6. At the end of the discussion, please indicate the strengths and weaknesses of the research conducted.
Regards!
Author Response
Reply to Comments from Reviewer 1
We are pleased to have this opportunity to revise our paper entitled, “Survey of Clinical Practice Patterns of Korean Medicine Doctors for Anorexia in Children: a Preliminary Study for Clinical Practice Guidelines (Manuscript ID children-1909169).” In revising the paper, we have carefully considered reviewers' comments and suggestions. As instructed, we have attempted to respond to all the reviewers’ comments.
Comment 1
- it is not clear what the purpose of the study is. Of course, I can see the purpose at the end of the theoretical introduction, but it needs to be detailed and provide hypotheses (research questions), and the introduction itself does not clearly present the research gap. I suggest before the objective to specify the assumptions of the study being conducted.
RESPONSE 1: We thank the reviewer for these helpful comments. As recommended by the reviewer, in order to further clarify the purpose of this study, we have added some sentences at the end of the introduction and modified as follows:
“Herbal medicines and acupuncture have shown significant effects among children having anorexia [21-23], but in either field of KM or Western medicine, CPGs have not been developed for anorexia in children in Korea. When it comes to treating children having anorexia with KM, diagnosis or treatment has not been standardized. Anorexia is recognized to be a major problem for young children under the age of three, but 7–27% of school-aged children also experience it [24]. In Korea, not only infants under 3 years old, but also children after school age are visiting KM institutions to improve their digestive problems including anorexia [12]. Therefore, comprehensive CPGs for anorexia treatment targeting not only infants but also all children under the age of 20 is necessary. The CPGs need to reflect the current status of KM treatment, but an investigation into the current status of the treatment methods used by KM doctors (KMDs) has not been made. This study aimed to investigate the diagnosis and treatment methods currently used by KMDs through a survey and to produce relevant CPGs.”
Comment 2
- please establish the criteria for qualifying the participants of the study.
RESPONSE 2: As the reviewer suggested, it has been modified as follows to make it more detailed and clearer.
“We conducted a survey targeting participants who satisfied the following conditions: KMDs who (1) graduated from the University of KM, (2) have KM licenses, (3) are members of the Association of Korean Medicine, and (4) have answered the questionnaire by September 17th.”
Comment 3
- I encourage you to expand statistical analyses to include the use of inference with statistical tests and determine the level of probability (p-value).
RESPONSE 3: We thank the reviewer for this insightful comment. As recommended by the reviewer, the P-values were calculated through a Chi-square test, and the following sentences were added to the method, results, and discussion:
“To compare by categorical data of clinical experience of the respondents, we analyzed the data with the Chi-square test. P-values < 0.05 were considered statistically significant.”
“When analyzing the usage rates of major KM treatment methods by clinical experience of the respondents, a statistically significant difference was found in the utilization rates of treatment between different experience groups, with the p-value of herbal medicine showing 0.016 and that of acupuncture < 0.001. In the case of moxibustion, there was no difference in the utilization rate between the five groups (p > 0.05) (Table 4).
Table 4. Utilization patterns of main treatment methods by clinical experience
|
Clinical experience |
|||||
Treatment |
≤4 years (n=57) (n (%)) |
5–9 years (n=109) (n (%)) |
10–19 years (n=118) (n (%)) |
20–29 years (n=76) (n (%)) |
≥30 years (n=24) (n (%)) |
p-value (Χ2 Test) |
Herbal medicine |
54 (94.7) |
106 (97.2) |
115 (97.5) |
76 (100) |
24 (100) |
0.016 |
Acupuncture |
37 (64.9) |
71 (65.1) |
65 (55.1) |
20 (26.3) |
18 (75) |
<0.001 |
Moxibustion |
26 (45.6) |
52 (47.7) |
50 (42.4) |
9 (11.8) |
16 (66.7) |
0.068 |
“When we analyzed the groups according to the length of their clinical experience, there was a statistically significant difference in the use of herbal medicine (p = 0.016) and acupuncture (p < 0.001) to treat anorexia in children (Chi-square test).”
Comment 4
- The methodology lacks mention of how many study participants there were - also refers to point 2 of this review, as this also refers to the inclusion and exclusion criteria of the study.
RESPONSE 4: As recommended by the reviewer, the inclusion criteria were clarified (RESPONSE 2), and we have added the following as exclusion criteria: "Participants who did not consent to participate in the study were excluded."
In addition, information on the total number of participants and the final number of participants is listed below in the Result (lines 134–136): “Of the 23,910 KMDs to whom the email was sent, 738 accessed the link and 385 responded to the survey. Except for one person who did not agree to participate, 384 KMDs completed the questionnaire, and the response rate was 1.61%.”
Comment 5
- In the titles of the tables, the abundance (N valid) should be given.
RESPONSE 5: As the reviewer had recommended, we put the N valid in the titles of the tables.
Comment 6
- At the end of the discussion, please indicate the strengths and weaknesses of the research conducted.
RESPONSE 6:
As advised by the reviewer, we added the following to demonstrate the weaknesses and strengths of the discussion.
“In this cross-sectional study, only a fraction of all KMDs completed the response, but it was possible to provide the general overview on the treatment of anorexia in children by KMDs, including various working regions, workplaces, ages, and clinical experience.”
Also, we have already described weaknesses in the discussion as follows:
“This study had several limitations. First, the response rate to the questionnaire was low. As the email was sent to all KMDs registered with the Association of Korean Medicine, there may have been invalid email addresses, and some KMDs may not have opened the email at all. Furthermore, the clinical status or opinions of KMDs who were not contacted by email were not reflected. Second, because the effectiveness and safety of the questionnaire were reported from the perspective of KMDs, a prospective study using actual patient data is needed to obtain more objective findings. In addition, to supplement the cross-sectional results of the email survey, in-depth interviews with KMDs, pediatric patients, or caregivers who have received KM treatment can be conducted. A third limitation is our use of a non-standardized questionnaire. However, to account for this, KM pediatric specialists participated in designing the questionnaire, and revisions were made based on the opinions of a panel of extramural KM experts.”
Overall, we deeply appreciate the reviewers’ constructive feedback on our original submission, which was very helpful. After addressing the issues raised, we believe the quality of the paper has much improved and hope you agree. We also hope that the revised version of our paper is now suitable for publication in Children and we look forward to your favorable response.
Yours sincerely,
Jihong Lee, KMD (The first author)
Department of Korean Pediatrics, College of Korean Medicine, Daegu Haany University, 136 Sinchendong–ro, Suseong‑gu, Daegu 42158, Republic of Korea.
Gyu Tae Chang, PhD, KMD (corresponding Author)
Department of Korean Pediatrics, College of Korean Medicine Kyung Hee University, Kyung Hee University Hospital at Gangdong, 892 Dongnam-ro, Gangdong-gu, Seoul 05278, Republic of Korea
Reviewer 2 Report
Dear Authors,
Thank you for your manuscript. Being a representative of Western medicine, it is hard for me to judge the scientific quality of the paper. I have only a few minor comments.
First, in the abstract, the Introduction and the Methods sections, it was never mentioned the age of children being treated for anorexia. Only in the Discussion section this information surprisingly appears for the first time: "Therefore, we asked KMDs their opinion on extending the above criteria to all children aged 0–19 years" (lines 255-256). Also, in the Introduction section, possible reasons for childhood anorexia should be discussed in a more detailed way as they are age-specific. For example, in older children and adolescents, a significant role in developing anorexia might play body image-related factors such as body dissatisfaction and low self-esteem, tending to disordered eating behaviours.
Next, despite the entire questionnaire being attached to the supplementary files, in the Methods section, parts of the questionnaire should be described, and the factors (questions) investigated should be listed.
Finally, as the paper is submitted to the international journal, the main differences in the treatment of anorexia in KM and Western medicine should be discussed. For example, individual and group psychotherapy was never mentioned.
All the best.
Author Response
Reply to Comments from Reviewer 2
We are pleased to have this opportunity to revise our paper entitled, “Survey of Clinical Practice Patterns of Korean Medicine Doctors for Anorexia in Children: a Preliminary Study for Clinical Practice Guidelines (Manuscript ID children-1909169).” In revising the paper, we have carefully considered reviewers' comments and suggestions. As instructed, we have attempted to respond to all the reviewers’ comments.
Comment 1
First, in the abstract, the Introduction and the Methods sections, it was never mentioned the age of children being treated for anorexia. Only in the Discussion section this information surprisingly appears for the first time: "Therefore, we asked KMDs their opinion on extending the above criteria to all children aged 0–19 years" (lines 255-256). Also, in the Introduction section, possible reasons for childhood anorexia should be discussed in a more detailed way as they are age-specific. For example, in older children and adolescents, a significant role in developing anorexia might play body image-related factors such as body dissatisfaction and low self-esteem, tending to disordered eating behaviours.
RESPONSE 1: We thank the reviewer for these helpful comments. We added a description of the age to the introduction section as follows.
“and these criteria apply to children up to 3 years of age.”
“Anorexia is recognized to be a major problem for young children under the age of three, but 7–27% of school-aged children also experience it [24]. In Korea, not only infants under 3 years old, but also children after school age are visiting KM institutions to improve their digestive problems including anorexia [12]. Therefore, comprehensive CPGs for anorexia treatment targeting not only infants but also all children under the age of 20 is necessary.”
Comment 2
Next, despite the entire questionnaire being attached to the supplementary files, in the Methods section, parts of the questionnaire should be described, and the factors (questions) investigated should be listed.
RESPONSE 2: We thank the reviewer for these helpful comments. We modified the description of the final version of the questionnaire as recommended by the reviewers as follows: “The final version of the questionnaire included items on (a) the sociodemographic status of respondents (i.e., sex, age, year of clinical experience, affiliated institution, and subspecialty); (b) current status of diagnosis and KM treatment; (c) perception, safety, and effectiveness of KM treatment for anorexia in children; (d) the need for each CPG item.”
Comment 3
Finally, as the paper is submitted to the international journal, the main differences in the treatment of anorexia in KM and Western medicine should be discussed. For example, individual and group psychotherapy was never mentioned.
RESPONSE 3: We thank the reviewer for these useful comments. As recommended by the reviewer, we have added a sentence about the Western medicine treatment as follows:
“On the other hand, the treatment of anorexia for children in Western medicine includes psychological treatment, zinc supplementation, and pharmacotherapy.”
In addition, a sentence on the development of Western medicine CPG for anorexia in Korea was added.
“Herbal medicines and acupuncture for anorexia in children have shown significant effects [21-23], but in either field of KM or Western medicine, CPGs have not been developed for anorexia in children in Korea.”
Once again, thank you for the helpful and constructive comments on the original version of our manuscript.
Yours sincerely,
Jihong Lee, PhD, KMD (The first author)
Department of Korean Pediatrics, College of Korean Medicine, Daegu Haany University, 136 Sinchendong–ro, Suseong‑gu, Daegu 42158, Republic of Korea.
Gyu Tae Chang, PhD, KMD (corresponding Author)
Department of Korean Pediatrics, College of Korean Medicine Kyung Hee University, Kyung Hee University Hospital at Gangdong, 892 Dongnam-ro, Gangdong-gu, Seoul 05278, Republic of Korea
Round 2
Reviewer 1 Report
Thank you for the changes made.